# Antioxidant and Antitumor Activities of Newly Synthesized Hesperetin Derivatives

**DOI:** 10.3390/molecules27030879

**Published:** 2022-01-27

**Authors:** Guanlin Zhong, Jiayi Shen, Zhengwang Chen, Zunxian Lin, Lipeng Long, Jiaying Wu, Chenhuan Long, Siyu Huang, Ping Lian, Guotian Luo

**Affiliations:** Key Laboratory of Organo-Pharmaceutical Chemistry of Jiangxi Province, Gannan Normal University, Ganzhou 341000, China; z01234560327@163.com (G.Z.); chenzwang@126.com (Z.C.); linzunxiangnnu@126.com (Z.L.); long_lipeng@126.com (L.L.); realwujiaying@163.com (J.W.); LCH1809105833@163.com (C.L.); JRL336699@163.com (S.H.); lianping@gnnu.edu.cn (P.L.)

**Keywords:** hesperetin derivatives, aminobenzene methylation, antioxidant activity, antitumor activity, structure–activity relationships, molecular docking

## Abstract

Hesperetin is a class of natural products with a wide range of sources and remarkable biological activities. In this study, we described the synthesis of a series of novel hesperetin derivatives and evaluated the in vitro antioxidant and antitumor activity of these compounds. Eleven novel compounds were synthesized in moderate yields. The compounds synthesized in this work exhibited antioxidant activities against DPPH and ABTS free radicals in a dose-dependent manner. Among them, compound **3f** had the best antioxidant activity, with IC_50_ of 1.2 μM and 24 μM for DPPH and ABTS, respectively. The antitumor activity of the compounds against human cancer cell lines, such as breast MCF-7, liver HepG2, and cervical Hela, was determined by a standard 3-(4,5-dimethyl-2-thiazolyl)-2,5-diphenyl-2-*H*-tetrazolium bromide (MTT) assay. Three compounds had moderate IC_50_ values. Interestingly, compound **3f** had better biological activity than hesperetin, which matches the prediction by Maestro from Schrödinger. Therefore, the new hesperidin derivative is a promising drug for the treatment of cancer due to its effective antitumor activity. The results also suggested that the antitumor activities of hesperetin derivatives may be related to their antioxidant activities.

## 1. Introduction

Flavonoid compounds are a kind of natural product with a wide range of natural existence. Hesperidin and hesperetin are the main active ingredients of citrus fruits, and hesperetin is a natural product worthy of study in citrus [1]. Application research of the natural product hesperetin is a hot frontier in related neighborhoods [2]. Hesperetin has a variety of pharmacological activities, such as anti-inflammatory [3], antioxidant [4,5], antibacterial [6,7], anticancer [8,9], anti-hepatic fibrosis [10], and immune regulation [11]. However, since hesperidin and hesperetin are easily decomposed and absorbed in the gastrointestinal tract after oral administration [12], the bioavailability of hesperidin and hesperetin is low, and the activity is general [13]. Therefore, the structural modification of hesperetin to better achieve the biologic activity of hesperetin is worthy of study.

Previously, many studies have found that hesperetin has antitumor activity because it elicits apoptosis in many cancer cells [14,15,16]. Hesperetin has multiple phenol structures that may be replaced by all kinds of functional groups in the benzene ring. Existing research has revealed that structurally modified 7,3′-dimethoxy hesperetin induces apoptosis [17]. The hydrogen bond between 5-hydroxy of hesperetin and 4-ketochromate was also found to enhance the application ability of hesperetin [18]. In addition, in a study of improving anticancer activity, it was also found that the structurally modified hesperetin at 4′- and 7-positions had better anticancer activity [19]. It can be concluded that the biological activity depends to a large extent on the structural characteristics; however, its 6-position derivatives have rarely been reported with antitumor activities. It is well known that the changes in the biological activity of compounds are caused by minor structural modifications of the bulk structure of compounds. Because many active compounds are similar to alkaloids in Chinese and Western medicine, the introduction of bases to compounds can improve their biological activity [20]. As we all know, benzyl amine derivatives widely exist in natural products, and they are essential for medical compounds [21]. Thus, we want to bind the benzyl amines to the 6-position of hesperetin. By molecular docking, we found that protein molecules can provide space for the splicing of two structural frameworks. By optimizing the reaction temperature and reaction time, a higher C-6 regioselectivity can be obtained [22]. Moreover, hesperetin, as a flavonoid, has been proven to be one of the sources of natural antioxidants. Structural modification of hesperetin to find compounds with more antioxidant capacity for application in food and pharmaceutical industries will have more interesting application foregrounds. Hesperetin can be used as natural antioxidants in the food and medical industries with broad application prospects [23,24]. Therefore, in this work, we report the synthesis of new hesperetin derivatives and evaluate the antioxidant and antitumor activities of these compounds. Our team looks forward to finding compounds that can simultaneously enhance a variety of biological activities in these hesperetin derivatives and to promoting further application development.

## 2. Results

### 2.1. Chemistry

Hesperetin derivatives (**3a**–**3k**) are obtained by the steps described in Figure 1. Firstly, hesperidin **1** was hydrolyzed with sulfuric acid in methanol to obtain hesperidin **2**, and then the target compound was synthesized. The synthesis strategy of hesperetin derivatives is the benzylation of hesperetin with benzaldehyde and amine. In our experiment, hesperetin derivatives were synthesized at the C-6 position, which were amine benzyl derivatives (**3a**–**3k**), by an electrophilic substitution reaction in methanol at 40 °C.

The scope of this three-component reaction was investigated. As presented in Figure 1, a wide range of aliphatic and aromatic amines were tolerated to furnish the substituted hesperetin **2**. Straight-chain, cyclic, primary, and secondary amines were all reactive in this multicomponent reaction (**3a**–**3d**). The secondary aryl alkyl amine afforded **3e** in 55% yield. Benzyl amines reacted efficiently to render the products **3f** and **3g** in 47% and 66% yields, respectively. Aromatic amines bearing either an electron-rich or an electron-poor group on the benzene ring were compatible and delivered the products in moderate yields (**3j**, **3k**). It can be clearly seen from the ^1^H-MNR of **3a** that the proton at the C-6 position of hesperetin disappeared. The structure of the ultimate product was determined by ^1^H-NMR, ^13^C-NMR, and HRMS spectral characterization data.

### 2.2. Antioxidant Activities Research In Vitro

It is reported that natural plant products can scavenge free radicals that result in cell death, histological damage, and chronic conditions development [25]. Hesperetin has antioxidant activity. Restraining or eliminating the priming or reproduction of superfluous active substances can postpone or inhibit cell injury caused by physiological oxidants and decrease the risk of potential human health influence related to oxidative stress or free radicals [23,26]. It is significant to measure the antioxidant activity of hesperetin derivatives in the exploration of their potential applications in the food, pharmaceutical, or cosmetics industries. For different antioxidant mechanisms, many methods for testing the antioxidant activity of compounds have been developed [24]. In this study, DPPH and ABTS testing methods [27] were selected for antioxidant research. The mechanisms of both methods are related to electron transport and the revivification of colored oxygenant; this makes it easy to observe the antioxidant activity of the tested compounds through the change of absorbance. Vitamin C (V_C_) was a positive control. IC_50_ values were computed using IBM SPSS Statistics 25.0.

As shown in Table 1, vitamin C as a positive control showed strong antioxidant activity. Obviously, compared with hesperetin (70 μM and 276 μM in DPPH and ABTS assays, respectively), the IC_50_ of all hesperetin derivatives were significantly lower, and some compounds were lower than those of V_C_ (59 μM and 236 μM in DPPH and ABTS assays, respectively). It is gratifying that compound **3f** showed the best antioxidant capacity, the IC_50_ value in the DPPH method was 1.2 μM, and the IC_50_ value in the ABTS method was 24 μM. It can be seen that the antioxidant capacity of the compounds was markedly improved by the structural modification of hesperetin.

### 2.3. Molecular Docking

We simulated the molecular docking between hesperetin and carcinogenic protein molecules to intuitively describe the reason why this structural modification improves the antitumor ability. As a crucial molecule of HPV16 carcinogenesis, the E6 protein is an important transformation protein for cell proliferation [28]. The outcomes are demonstrated in Figure 1. The docking results indicate hesperetin **2** interacted with E6 protein (Figure 1a). Compound **3f** interacts with the E6 protein has a better Glide score of −5.061 than hesperetin **2** (Figure 1b), which is −4.639.

### 2.4. Antitumor Activity Research In Vitro

To explore the anticancer activity of the synthetic compounds, our team studied the antitumor activity of hesperetin and its derivatives in MCF-7, HepG2, and HeLa cells. Compound concentration was set at 50 μM for screening; screening found that several hesperetin derivatives showed strong antiproliferative activity, as shown in Table 2. Seven compounds **2**, **3a**, **3e**, **3f**, **3g**, **3i**, and **3k**, exhibited preferable antiproliferative activity against one or more cancer cells (inhibition rates were greater than 50%), which could be further used for the determination of IC_50_ values.

As shown in Table 3, through the calculation of IC_50_, with cisplatin as the positive control, the IC_50_ values of the screened compounds on MCF-7, HepG2, and HeLa cells were higher than those of cisplatin, and the cytotoxicity of the compounds was lower than that of cisplatin. Furthermore, compared with hesperidin, compounds **3e**, **3f**, and **3k**, through structural modification, showed significant antiproliferative activities against the three cancer cells. Among them, compound 3**f** had the strongest antiproliferative activity, and its IC_50_ values for the three cancer cells were 5.3 μM, 8.8 μM, and 8.6 μM, respectively.

As shown in Appendix A, the inhibition rate of hesperetin **2** and its derivatives **3e**, **3f**, **3k**, and cisplatin on three kinds of cancer cells, MCF-7, HepG2, and HeLa cells, varies with the strength of the solution. With different strength of solutions (0.01, 0.1, 1, 10, 50 μM) of hesperetin **2** and its derivatives **3e**, **3f**, **3k** and cisplatin on MCF-7, HepG2, and HeLa cells, the inhibition rate of cancer cells increases with the increase in concentration, and the differences of different compounds are compared. Compared with hesperetin, its derivatives have a stronger inhibitory effect on three kinds of cancer cells in a dose-dependent manner.

## 3. Discussion

In the reaction in our work, the benzaldehyde protonates to the carbonyl group, and the amine compound undergoes nucleophilic addition to the carbonyl group, deprotonation, electron transfer on the nitrogen, and water departure to obtain an electrophilic imine ion intermediate. The intermediate attacks the active hydrogen at the C-6 position of the hesperetin A-ring, loses the proton, and obtains the final product. This method is widely used in the synthesis and modification of our lead compounds.

As shown in Table 1, it is worth noting that almost all hesperetin derivatives have stronger activities than hesperetin in terms of antioxidative capacity appraised by DPPH and ABTS methods. Compared with V_C_, the activity of compounds evaluated by the DPPH method was almost better than that of V_C_, and the activity of only half of hesperetin derivatives was stronger than that of V_C_ evaluated by the ABTS method. All compounds cleared free radicals in a dose-dependent manner. This confirmed that antioxidant activity displayed by structurally modified derivatives was greatly improved in vitro. Hesperetin derivatives are expected to become antioxidants.

As shown in Table 2, it is worth noting that for HeLa cancer cell lines, almost all hesperetin derivatives are more potent than hesperetin, while for HepG2 cancer cell lines, only half of hesperetin derivatives have stronger activity than compounds **2**. Appendix A confirms that in vitro, the antitumor of the structurally modified derivatives is greatly improved. Interestingly, the inhibitory ability displayed by derivatives on MCF-7 and HeLa is better than HepG2, indicating that MCF-7 and HeLa are more sensitive to derivatives than HepG2. In a word, the structural modification of this series of hesperetin derivatives is successful. The results also suggested that the antitumor activities of hesperetin derivatives may be related to their antioxidant activities [29].

In order to explore the binding space between E6 protein and the compounds, we initially introduced some aliphatic chains at the 6-position of the structure, so we synthesized **3a**, **3b**, **3c**, and **3d**. The activities of **3a**, **3b**, **3c**, and **3d** are similar to compound **2**, suggesting that their antiproliferative activity is not affected by fatty amines. We noticed that the introduction of the aliphatic chain has little effect on the potency, so we introduced a larger benzene ring and synthesized the compounds **3e**, **3f**, and **3g**, and found that the activity was improved. Subsequently, we synthesized the derivatives with two aromatic rings as **3h**, and found that the activity decreased. So, we kept the strategy of introducing an aromatic ring at the 6- position, changed the position of substituents of the benzene ring, and found that keeping a group at the para position of the benzene ring was beneficial to the potency. For example, compound **3k** has good inhibitory activity. As shown in Figure 2, we draw a schematic diagram of the structure-activity relationship of compound **3f**, according to our designed experimental results. This confirms that in vitro antitumor activities displayed by structurally modified derivatives are greatly improved. Interestingly, the inhibitory ability displayed by derivatives on MCF-7 and HeLa is better than HepG2, indicating that MCF-7 and HeLa are more sensitive to derivatives than HepG2.

The molecular docking in our work also indicates that hesperetin **2** has an obvious binding affinity to E6 protein, and hesperetin **2** binds within the active area of E6 protein. This means that structural modifications can be made to improve binding affinity. Furthermore, we noticed that between the phenyl groups of Trp2 and **3f**, there is a typical π-π conjugation. The amino group established hydrogen bonds with the Ser103 and the Afc105. These interactions explain why compound **3f** has more potency than other compounds. The docking results show that this is consistent with the activity of compound **3f**.

## 4. Materials and Methods

### 4.1. Chemistry

#### 4.1.1. Synthesis of Hesperetin (**2**)

Hydrolysis of hesperidin to hesperetin is generally easy to implement. We used about 3.5 g. 5.73 mmol of hesperidin **1**, adding 280 mL methanol and 10 mL 96% H_2_SO_4_ while stirring. The reaction solution was set to heating temperature 70 °C and refluxed for 7.5 h. After the reaction stopped heating, the reaction mixture was cooled to room temperature. Then, the reaction solution was extracted and poured into EtOAc about 300 mL. The organic phase was washed with water (3 × 400 mL) until the lower aqueous phase was colorless and transparent, and then the upper organic phase was concentrated. Then, acetone was used to dissolve the residue about 70 mL and drop it into the hot 1% AcOH aqueous solution of about 700 mL. The heating temperature was set at 70 °C, and the stirring was 1.5. After solution cooling, filtration, and drying, hesperetin was obtained.

#### 4.1.2. Synthesis of Hesperetin Derivatives (**3a**–**3k**)

The synthesis strategy of hesperetin derivatives is the benzylation of hesperetin with benzaldehyde and amine. Briefly, 0.604 g of hesperetin (2 mmol) was taken in the reaction vessel, 40 mL of methanol was added to dissolve it. Then, we added 204 μL of benzaldehyde (2 mmo1, 1.0 equivalent) and 1.1 equivalent of the amine-containing compound; the reaction temperature was 40 °C. The reaction was completed after stirring for 8 h; then, the reaction was monitored by TLC. The reaction mixture was concentrated and evaporated in vacuum to obtain crude products. The reaction mixture was concentrated and evaporated in vacuum and extracted with ethyl acetate-water to a colorless and transparent aqueous phase. The organic layer of ethyl acetate was retained and dried with anhydrous Na_2_SO_4_ for 4 h. The mixture was concentrated under reduced pressure to obtainrough stuff, and the rough stuff was purified by silica gel column chromatography (EtOAc/PE, 3:1→1:1) to obtain compounds (**3****a**–**3****k**).

#### 4.1.3. Structure Characterization of Hesperetin and Hesperetin Derivatives (**3a**–**3k**)

The ^1^H-NMR and ^13^C-NMR spectra were obtained in CDCl_3_ on a 400 and 100 MHz NMR spectrometer (Bruker, Karlsruhe, Germany). The chemical shifts were referenced to signals at 7.26 and 77.0 ppm, respectively, and chloroform was used as the solvent with TMS as the internal standard. HRMS data were collected from a high-resolution mass spectrometer (QTOF-LCMS-9030, SHIMADZU) (Kyoto, Japan).

### 4.2. Antitumor Activities

#### 4.2.1. Cell Culture

Human MCF-7 cancer cells, HepG2 cancer cells, and Hela cancer cells were collected from Shanghai Cell Bank of Chinese Academy of Sciences (Shanghai, China). The frozen Human MCF-7 cancer cells, HepG2 cancer cells, and Hela cancer cells were quickly removed from the liquid nitrogen tank and shaken in a 37 °C water bath. After all the substances in the freezer were completely melted, the solution in the freezer was slowly dropped into a 10 mL centrifuge tube. The centrifuge tube had an appropriate amount of fresh medium (Gibco-BRL, Gaithersburg, MD, USA) pre-added to seal the membrane. At room temperature, we centrifuged at 1000× *g* for 5 min. We removed supernatant, suspended in 2 mL fresh culture RPMI1640, and transferred it to sterile culture bottle. We added the culture medium into 5–6 mL and cultured at 37 °C, 5% CO_2_ incubator. We replaced fresh medium 24 h later.

#### 4.2.2. MTT Assay

This study is of the cytotoxicity of hesperetin and hesperetin derivatives (**3a**–**3k**) to MCF-7, HepG2, and Hela cell lines by methyl thiazolyl tetrazolium (MTT) assays. First of all, MCF-7, HepG2, and Hela cell lines were seeded into a 96-well cell culture plate. The seed concentration was about 1 × 10^4^/hole; then, the cell lines were cultured at 37 °C under 5% CO_2_ for 24 h. After that, the concentration of hesperetin and hesperetin derivatives (**3a**–**3k**) (0, 0.01, 0.1, 1, 10, 50 μM) were added in these wells, respectively. After being incubated for another 48 h at the same conditions, the MTT solutions (10 µL, 0.5 mg/mL) were added to these wells. Then, cells were incubated for 4 h. The medium was removed, and 100 μL dimethyl sulfoxide was added to dissolve the formazan crystals. The OD570 was measured by an ELx800 microplate reader (Bio-Tek, Winooski, VT, USA) with subtraction of background absorbance. The cell viability was computed using following equation:Cell inhibition ratio (%) = (A_C_ − A_S_)/A_C_ × 100(1)
where A_C_ is the absorbance of the control, and A_S_ is the absorbance of dyes. All samples were analyzed in triplicate, and the results are expressed as mean ± standard deviation. The cytotoxicity of hesperetin and hesperetin derivatives (**3a**–**3k**) was expressed as an IC_50_, defined as the concentration causing a 50% reduction in cell growth compared with untreated cells.

#### 4.2.3. Molecular Docking

Docking study of hesperetin and hesperetin derivative **3****f** was carried out by Schrödinger’s Maestro and PyMOL programs. The crystal structure of HPV16 protein E6 (ID: 4XR8) was retrieved for this work from the protein database (PDB). The protein molecules were pretreated and stored in. pdbqt format. The B3LYP/6-311G ** level of density functional theory (DFT) was selected to predict the simulation, and the ligand molecules were optimized by Gaussian 03 software. In addition, ligands (isomers) such as hesperetin and compound **3f** were also saved in pdbqt format. In the docking studies using software, the grid box was set to 60 × 60 × 60 dimensions to select the protein molecules and ligand molecules, and the La-marckian genetic algorithm was used to find the global optimal conformation. Finally, the interaction between protein and ligand was evaluated by Glide score.

### 4.3. Free-Radical-Scavenging Capacity

#### 4.3.1. DPPH Antioxidant Activity Assay

In this study, stable and convenient free radical 2,2-diphenyl-1-picrylhydrazyl (DPPH) method was used to determine the free-radical-scavenging activity of hesperetin and its derivatives in this study. The experimental samples were diluted with ethanol into different concentrations of sample solution, and then the accurate amount of 0.5 mL sample solution and 2.5 mL 60 μM DPPH solution were dissolved in ethanol. The mixed solution was oscillated vigorously in the dark for 1 h, and the absorbance of the reaction solution was measured by UV spectrophotometer at 517 nm. V_C_ was positive control in this experiment. The expression formula of DPPH scavenging activity is as follows:DPPH scavenging activity (%) = (A_C_ − A_S_)/A_C_ × 100(2)
where A_C_ was the absorbance of negative control DPPH solution and As was the absorbance of 0.5 mL sample solution and 2.5 mL DPPH solution mixed solution. All the experimental samples were analyzed in three copies, and the results were expressed as mean ± standard deviation. The scavenging activity was expressed as 50% inhibitory concentration (IC_50_), namely, the sample concentration of 50% DPPH free radical scavenging after incubation.

#### 4.3.2. ABTS Antioxidant Activity Assay

The free-radical-scavenging ability of hesperetin and its derivatives was evaluated by stable and convenient ABTS method. The experimental samples were diluted with ethanol in turn into different concentrations of sample solutions, and then accurate amount was configured with 7 mM ABTS solution and 2.45 mM potassium persulfate solution with deionized water to produce ABTS radicals at room temperature in dark environment for 16 h. The absorbance of the accurate volume of ABTS solution diluted with ethanol to 734 nm was 0.70 ± 0.05. The diluted ABTS solution (2 mL) was mixed with 100 µL of the experimental sample, and the mixture was kept in dark at room temperature for 6 min. The absorbance of the mixed solution was measured by UV spectrophotometer at 734 nm. V_C_ was used as the positive control in this experiment. The expression formula of ABTS scavenging activity is as follows:ABTS scavenging activity (%) = (A_C_ − A_S_)/A_C_ × 100(3)
where A_C_ is the absorbance of the ABTS solution as the control and A_S_ is the absorbance of the ABTS solution after adding the sample. The IC_50_ was computed from the graph of scavenging percentage against concentration using IBM SPSS Statistics 25.0. The results are expressed as mean ± standard deviation of three experiments.

### 4.4. Statistical Analysis

All experiments were performed at least in triplicate, and each experiment was independently performed at least 3 times. The graphical presentations were performed using Origin 2018 64 Bit. Data were presented as the mean ± standard and were analyzed using SPSS 25.0 software (Chicago, IL, USA). Analysis of variance (ANOVA) for samples was determined by Tukey’s multiple range tests (*p* < 0.05).

## 5. Conclusions

In summary, a series of aminobenzylated hesperetin derivatives **3a**–**3k** were successfully synthesized, and their antioxidant activity and antiproliferative activity were evaluated. In this series, the antioxidant activity of almost all compounds was higher than that of hesperetin. Compound **3f** has the strongest antioxidant activity, and all compounds scavenged DPPH and ABTS free radicals in a dose-dependent manner. Meanwhile, the antiproliferative activity against three different cancer cells on MCF-7 cells, HepG2 cells, and HeLa cells of all the compounds is higher than that of hesperetin. Compound **3f** also has the strongest antiproliferative activity. In a molecular-docking study, compared with hesperetin, compound **3f** has a better interaction with the E6 protein, and the Glide score is also better. The results signified that the structural modification of hesperetin in the successful evaluation of derivatives was improved, further research is underway, and the results will be disclosed in due course.

## Data Availability

The data presented in this study are available in this article or the Appendix A.

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
