# Peer review of "Antioxidant and Antitumor Activities of Newly Synthesized Hesperetin Derivatives"

_molecules, 2022, doi:10.3390/molecules27030879_

Round 1

Reviewer 1 Report

The present paper reports the synthesis and assessment of antioxidant and antitumor activities of new hesperetin derivatives.

Twelve novel molecules were obtained in moderate yields. Compounds were characterized by 1H NMR, 13C NMR and HRMS data. Antitumour activity of hesperetin and its derivatives was evaluated in three cancer (MCF-7, HepG2 and HeLa) cell lines. Antioxidant activities were tested in DPPH and ABTS assays.

Comments:

According to the title of the paper, the antioxidant activity should be described before the antitumor activity.

There is no information about cisplatin in paragraph 4.2. Were the IC50 values for cisplatin in Table 2 determined in the same study? If so, why are the data for cisplatin not provided in Table 1 and Figure 3?

Antitumor activity of hesperetin and its derivatives should be described mainly in comparison to the commonly used anticancer drug cisplatin. Unfortunately, compared to cisplatin, none of the compounds tested were interesting in terms of anticancer activity.

The cytotoxicity of compounds against a normal cell line should be studied.

Hesperetin, its derivatives and vitamin C have different molar masses. Therefore, the antioxidant activity should be tested in micromolar concentrations and not in mg/ml.

Any linguistic and typing errors should be corrected, e.g. μnol/L, hesperectin, antioxygenic.

References should be carefully checked and adapted to the journal's requirements.

The language should be carefully modified.

Reviewer 2 Report

This manuscript describes interesting results on biological activities of flavanone derivatives, but the data presentation needs significant improvement, as outlined below.

  1. There are problems with NMR spectroscopy data of the synthesized compounds in Supplementary Material. In many cases, solvent peaks are the highest peaks in NMR, which makes the compound peaks hard to see. In such cases, the scale of NMR spectra should be changed to allow a better view of the compound peaks. It would be useful to assign the peaks, particularly in 13C NMR spectra, to prove the purity of the synthesized compounds. For instance, the 13C NMR spectrum of 3k contains many more peaks than that of 3j, despite their very similar chemical structures.
  2. The mass spectrometry data in Supplementary Material should include the experimental and simulated isotopic distributions of the target peak, to prove that the assigned molecular formula is correct. In some cases, the target peak is not the main peak in mass spectra (for 3b, 3g, 3h, 3j and 3k), or even not found at all (3l). It is unclear why the negative ion mode ([M-H]-) was used for mass spectrometry. The synthesized compounds should give good signals in the positive ion mode, due to protonation of the amine groups, and the positive ion mode is usually more sensitive. Details of the equipment and conditions used for mass spectrometry should be given in Supplementary Material.
  3. The choice of E6 protein for molecular docking studies looks random. Is there any literature data on the link between anti-cancer activities of small molecules, particularly flavanones, and their binding to E6 protein? Flavanone derivatives are better known as ion channel blockers (e.g., Straub et al. Pharmacol. 2013, 84, 736-750 or Alvarez-Collazo et al. Br. J. Pharmacol. 2019, 176, 1090-1105), why not use some of these proteins for docking studies?
  4. What are the likely reasons for a positive correlation between the antioxidant activity and cytotoxicity of the compound 3f? One would expect that high antioxidant activity leads to protective, rather than toxic, effect on cultured cells.
  5. The results of antioxidant activity studies in Table 3 should be presented in µM values, rather than in µg/mL, to match the cytotoxicity data in Tables 1 and 2. The titles of Tables 1 and 2 should include the treatment time used in cytotoxicity studies (48 h). The data in all three tables should use correct numbers of significant figures, such as 26 ± 6 instead of 25.6 ±1.
  6. The claim that ‘the IC50 values of the screened compounds were higher than those of cisplatin’ (first paragraph in p. 5) is misleading because it actually means that the compounds are less toxic than cisplatin.
  7. Presentation of the cytotoxicity data in Figure 3 uses an incorrect X scale (50 µM is out of place in the logarithmic scale). The size of the numbers and labelling on the axes used in Figure 3 should be increased. The corrected Figure 3 can be moved to Supplementary Material, because it doubles on the data contained in Table 2.
  8. Figure 1 and Schemes 1 and 2 contain duplicating material. Figure 1 can be removed, and the two Schemes combined into one.
  9. The first paragraph of the Introduction is too general and can be removed.
  10. The manuscript needs significant English editing, particularly the Material and Methods section. This section is written like a laboratory protocol, not a journal article.

Round 2

Reviewer 1 Report

The manuscript has been revised in accordance with my comments and suggestions.

Author Response

Response to Reviewer 1 Comments

Point 1: The manuscript has been revised in accordance with my comments and suggestions.

Response 1: Thank you very much for your comments and suggestions.

Reviewer 2 Report

I agree with most changes in the revised version, except for the following:

(1) the parent compound is correctly called hesperetin, not hesperectin.

(2) on p. S1 (Supplementary Material), the instrumental parameters used for mass spectrometry (e.g., temperature, cone voltage, sheath gas flow rate) should be given, rather than general notes about the collection of mass spectra.

(3) The MTT data on pp. 22-23 (Supplementary Material) still have an incorrect X scale. This figure should probably be called Figure S1, rather than Figure 1. 
